# Behavioural Improvements in Children with Autism Spectrum Disorder after Participation in an Adapted Judo Programme Followed by Deleterious Effects during the COVID-19 Lockdown

**DOI:** 10.3390/ijerph18168515

**Published:** 2021-08-12

**Authors:** Jose Morales, David H. Fukuda, Vanessa Garcia, Emanuela Pierantozzi, Cristina Curto, Josep O. Martínez-Ferrer, Antonia M. Gómez, Eduardo Carballeira, Myriam Guerra-Balic

**Affiliations:** 1Faculty of Psychology, Education Sciences and Sport Blanquerna, Ramon Llull University, 08022 Barcelona, Spain; vanessagn@blanquerna.url.edu (V.G.); cristinacl5@blanquerna.url.edu (C.C.); joseoriolmf@blanquerna.url.edu (J.O.M.-F.); antoniamariagh@blanquerna.url.edu (A.M.G.); MiriamElisaGB@blanquerna.url.edu (M.G.-B.); 2School of Kinesiology & Physical Therapy, University of Central Florida, Orlando, FL 32816, USA; David.Fukuda@ucf.edu; 3School of Exercise and Sport Sciences, University of Genoa, 17100 Genoa, Italy; emanuela.pierantozzi@unige.it; 4Department of Physical Education and Sport, University of A Coruna, 15179 Oleiros, Spain; eduardo.carballeira@udc.es

**Keywords:** ASD, Autism, adapted judo programme, exercise intervention, physical activity, COVID-19, lockdown, GARS

## Abstract

The public health lockdown prompted by the novel coronavirus (COVID-19) pandemic, which included school closures that may have potentially serious consequences for people with disabilities or special educational needs, disrupted an ongoing adapted judo training intervention in children with Autism Spectrum Disorder (ASD). The purpose of this study was to compare repetitive behaviours, social interaction, social communication, emotional responses, cognitive style and maladaptive speech scores across four time-points: baseline, after an eight-week control period, after an eight-week judo intervention and after an eight-week lockdown period due to COVID-19. The sample consisted of 11 children diagnosed with ASD according to the criteria of the Diagnostic and Statistical Manual of Mental Disorders—Fifth Edition (DSM-V), with an intelligence quotient (IQ) range between 60 and 70. Significant improvements were shown following the judo intervention period compared to the baseline and control periods. However, the same values significantly declined during the COVID-19 lockdown period resulting in values lower than those recorded at baseline, and following the control period and the judo intervention. The decline in psychosocial and behavioural scores are likely due to the stress caused by the sudden halt in activity and the increase in sedentary practices associated with the lockdown.

## 1. Introduction

Autism Spectrum Disorder (ASD) is a developmental disorder that involves deficits in social interaction, communication and behaviour. Children with ASD are at risk of physical inactivity due to social and behavioural problems [1]. These individuals tend to spend less time on physical exercise [2] while displaying more deficient motor skills and physical conditioning than their typically developing children [3,4]. In recent years, researchers have attempted to quantify the effects of physical exercise on the motor skills of children with ASD and to develop recommendations for professionals who work with this population [5,6]. The potential for physical activity to enhance the relationship and communication skills in children with ASD has been established, with evidence showing that exercise can lead to improvements in social interactions with classmates, parents, siblings and teachers [7,8,9].

Despite the clear developmental benefits of physical activity, many children with ASD have a relatively sedentary lifestyle [10]. Short-term participation in physical activity could lead to a decrease in functional difficulties, including behaviours like hyperactivity, aggression and self-harm that are common in children with ASD [1]. In addition, long-term engagement may reduce mortality and morbidity associated with chronic adult diseases such as cardiovascular dysfunction and obesity [1,10]. Unfortunately, individuals with ASD often face obstacles that limit engagement in physical activity levels under normal circumstances. Participating in sports is often further limited because of behavioural problems, motor skill deficits or a lack of trained instructors or peer exercise partners [11].

In response to these challenges, several programmes aimed exclusively at people with ASD have been developed, including a range of physical activities and sports initiatives to enhance social skills and quality of life. The results of these investigations have been positive, with reports of improved social, communication, self-regulation, and motor skill [12,13,14,15]. The benefits appear to be more than transient changes, as demonstrated by Zanobini and Solari [16], who found improved relational behaviours and aquatic skills six months after participating in a swimming programme. Therefore, additional information on successful physical activity engagement for children with ASD and promotion of the enjoyment of sports may assist in limiting the potentially harmful effects of a sedentary lifestyle within this population.

The available evidence examining the use of adapted martial arts activities for individuals with ASD shows that participation can be effective, especially when it comes to enhancing motor skills [17,18] and social behaviour. For example, a study of karate participants reported significant improvements in stereotypical behaviour and social interaction [19,20]. While martial arts training may be beneficial due to coupling moderate to high-intensity physical exercise with mental skills or mindfulness practice [21], it may be particularly appealing to those with ASD because of the repetitive structure of the movements involved [22]. Aikido has been found to reduce the symptoms of children with ASD associated with social ability, physical ability and communication behaviours [23]. Recent studies designed to study the effects of participation in judo on children with ASD have yielded initial results that point toward some psychosocial improvements [24]. These programmes also seem to represent an effective way to promote moderate to high-intensity physical activity among this population and curb their tendency toward a sedentary lifestyle [21]. We believe that judo practice, and the overall health benefits it offers like other forms of physical exercise, also provides specific benefits that come from the opportunity to establish physical contact mediated through the grip in the judo uniform with their partners. It has been demonstrated that physical contact, competition and cooperation situations that occur in judo sessions contribute to the well-being and social integration of children in the general population [25,26] and children with intellectual disabilities in particular [27]. These benefits could include discipline, respect, cognitive aspects, autonomy, and not specifically physical fitness, but functional fitness that will improve daily life activities. All this together would facilitate children with ASD being part of society and enjoying being included in it.

The literature shows that the primary objective of most sports programmes aimed at people with ASD is to reduce the prevalence of sedentary behaviour [2]. Inactivity prevention is necessary because the patterns of communication difficulties, anxiety, and lack of social interaction typical of this population tend to be associated with low levels of participation in moderate to high-intensity physical activities and greater amounts of time spent engaging in sedentary behaviours [28]. In this sense, martial arts training can also be helpful to reduce stress and anxiety/depression symptoms in autistic children [29,30]. In particular, the practice of judo has shown positive results in typically developing adolescents [31].

Meanwhile, the extremely unusual situation that struck the world in 2020 due to the novel coronavirus (COVID-19) has caused school closings and confined children to their homes [32]. Consequently, all the adapted sport and physical activity programmes for children with ASD have been brought to a halt. The suspension of these activities will have potentially serious consequences for people with disabilities or special educational needs, as they are vulnerable to abandonment and lack of stimuli [33]. This unprecedented lockdown situation and the resulting lack of physical activity and excess of sedentary behaviour could have unexpected effects on children with ASD.

When the COVID-19 crisis began, we were in the midst of an adapted judo project with a group of children with ASD, with the backing of the European Union’s ERASMUS + Sport programme. The initiative included six different countries and was scheduled to run through the 2020–2021 and 2021–2022 school years. The original project proposal called for measuring the effects of an eight-week adapted judo programme on the children’s motor and psychosocial skills. The sudden interruption of the programme because of the COVID-19 health crisis and lockdown meant that we could not complete the second round of motor skill testing. We were, however, able to administer the post-test questionnaires on the psychosocial variables. More importantly, the situation afforded us the opportunity to take the research in an innovative direction by collecting psychosocial data for eight weeks of the lockdown and comparing these data to those collected during the intervention period and baseline values.

Thus, the main objective of this study is to compare the behavioural scores on the six Gilliam Autism Rating Scale (GARS) subscales obtained by children with ASD during three different periods (baseline/control, judo intervention and lockdown). We hypothesise that the participants will show improved behaviour during the adapted judo intervention and that their behaviour ratings will decline during the COVID-19 lockdown period. Due to the relatively short-term duration of the training intervention, social interaction and social communication were expected to exhibit the most pronounced improvements.

## 2. Materials and Methods

### 2.1. Participants

We recruited a convenience sample in special education schools that consisted of 11 boys (*n* = 7) and girls (*n* = 4) ranging from 9 to 13 years of age with an average age of 10.17 (±2.45) years, an average height of 153.18 (±6.48) cm and an average weight of 53.71 (±6.11) kg. Initially, the study began with 15 participants, 2 of them dropped out for reasons unrelated to the study, and 2 of them did not complete all the sessions. Finally, the data of the 11 participants were used (Figure 1). The children were invited to participate via several associations of families with children with ASD and special education schools from Barcelona (Spain) area. All participants had been diagnosed with ASD according to the criteria of the Diagnostic and Statistical Manual of Mental Disorders—Fifth Edition (DSM-V), and the psychological reports provided by the participants reported an intelligence quotient (IQ) range between 60 and 70 (mean of 66.5 ± 3.77). Individuals who had been advised against physical activity for medical reasons were excluded, as were those who had previously taken judo classes. The participants were invited to participate in the study voluntarily, and along with their parents, they were informed verbally and in writing as to the programme’s characteristics. Parents or legal guardians signed informed consent forms, and the children signed a consent document that explained the objectives and planned activities of the programme. The study was approved by the Research Ethics Committee of Ramon Llull University with reference number CER URL_2019_2020_003. All protocols applied in this research (including managing the participants’ personal data) complied with the requirements specified in the Declaration of Helsinki of 1975 and its subsequent revisions. The trial was registered in clinicaltrials.gov (NCT04523805).

### 2.2. Procedure

After the preliminary procedures described above, the research design consisted of a longitudinal study in which data were collected on each participant at four different time points (Figure 2) between 23 November 2019 and 9 May 2020. Baseline data were collected at time point 1 (T1-Baseline), at the start of the programme, and eight weeks later, scores were recorded for time point 2 (T2-Control), which represents a control period. During the control period, part of which included the winter break from classes, the students did not participate in any extracurricular physical activities; thus, their organised physical exercise was limited to their regular physical education classes at school. Between this second measurement and time point 3 (T3-Judo), the eight-week judo intervention was conducted, which consisted of weekly adapted 75-min judo sessions. The final period consisted of eight weeks of the obligatory lockdown imposed by the health authorities in Spain due to the COVID-19 crisis on 14 March 2020, culminating in the final measurement at time point 4 (T4-Lockdown).

The choice was made to include the control period at the start of the process because we did not have a control group willing to submit to all the measurements throughout the project. In the absence of a control group, it was decided to take the second measurement, after which the participants only took part in the compulsory physical education activities at their schools.

The same adapted Judo programme is carried out in six countries of the European Union; nevertheless, only data of one of the countries are included in the present study. The reasons behind that decision are the insufficient data homogeneity in all the countries based on varying lockdown dates, the lack of control period and different intervention times. Therefore, we considered it appropriate to include only the data from one of the countries to ensure rigour and control in the data collection as part of an initial evaluation before examining a larger sample.

### 2.3. Intervention

The judo sessions were performed in a large and well-ventilated space suitable for judo practice, such that the safety of the participants was maintained. The judo equipment required for this project included a tatami mat with a surface area of 120 m^2^, made of high-density foam that helps prevent injuries and ensures that a wide range of activities can be carried out safely. Each participant wore a judogi (a traditional uniform consisting of a cotton jacket and trousers and a belt).

The sessions were 75 min in duration and were held once a week. Two judo teachers with degrees in pedagogy and sports sciences and 7th and 6th-degree black belts, respectively, led each session, and at least four volunteer judo instructors were present to lend support. The sessions were divided into a warm-up, main exercise and cool-down activities. The main exercise content of the sessions included:Different types of movements and falling techniques (from walking in all directions to turning around, from stable movements to unstable movements).Judo techniques and opposition games (building up body contact with games, teaching simplified movements, basic judo movements).Ground control techniques and throws (gradually adding techniques to already known movements, scaffolding basic repetitive movements to assist in understanding those more relevant for judo).Repetition of different forms of foundational directional movements (pulling, pushing, holding, lifting).

The instructional methodology applied the principles of gradual progression, featuring practice to consolidate the concepts learned in the initial lessons before moving on to more complex material. Each participant was allowed to progress at their own pace. Learning was based on imitation and guided modelling of techniques.

### 2.4. Instruments

All participants were assessed at the previously described timepoints using the Gilliam Autism Rating Scale-Third Edition (GARS-3) scale [34]. The GARS-3 is one of the most commonly used instruments to assess changes in the severity of ASD behaviours. It includes 56 items describing the characteristic behaviours of individuals with ASD. The items consisted of six subscales: repetitive behaviours, social interaction, social communication, emotional responses, cognitive style, and maladaptive speech. Parents or caregivers scored each item on a four-point Likert-type scale (0 = never observed; to 3 = frequently observed). A higher score indicates severity of autism-related behaviours, and a lower value represents an improvement. The instrument can be administered in 5–10 min, and it is based on the frequency of occurrence of each item under ordinary circumstances in a six-hour period. The raw score for each subscale was used.

For the first three measurements (T1-Baseline, T2-Control and T3-Judo), parents completed the questionnaire with pen and paper, and they were allowed to ask questions about the interpretation of a given item. For the final measurement (T4-Lockdown), parents and caregivers received a Google Forms hyperlink by email and completed the GARS-3 online. Parents and caregivers did not report any issue when completing the online questionnaire, likely because they were already familiar with the instrument.

### 2.5. Statistical Analysis

All descriptive data from the dependent variables are presented as mean ± standard deviation (SD). The normal distribution of each variable was checked with a Shapiro–Wilk test. The analysis of the outputs from each GARS-3 subscale at the four-time points (T1-Baseline, T2-Control, T3-Judo and T4-Lockdown) were carried out using one-way repeated measures multivariate ANOVA with follow-up univariate analyses with Bonferroni post hoc correction for multiple comparisons. All statistical analyses were conducted using the Statistical Package for Social Science version 24.0 software (SPSS, Inc., Chicago, IL, USA). A significance level of *p* < 0.05 was used for all tests.

## 3. Results

One-way repeated measures multivariate ANOVA of the mean GARS-3 score across all subscales showed a significant main effect within subjects (F_18,81_ = 4.75, *p* < 0.05; η_p_^2^ = 0.51). Univariate contrast showed a significant effect for time on four of the six subscales: repetitive behaviours, social interaction, social communication and emotional responses. In all cases, the sphericity assumption was violated and the number of degrees of freedom was adjusted using the Huynh–Feldt method, repetitive behaviours (F_1.36,15.09_ = 15.48, *p* < 0.05; η_p_^2^ = 0.61); social interaction (F_1.12,11.62_ = 25.55, *p* < 0.05; η_p_^2^ = 0.71); social communication (F_1.27,13.82_ = 18.21, *p* < 0.05; η_p_^2^ = 0.64); emotional responses (F_1.68,19.89_ = 76.95, *p* < 0.05; η_p_^2^ = 0.88).

Pairwise comparisons (Figure 3) indicated a significant improvement (*p* < 0.05) following the 8-week adapted judo training intervention at T3-Judo in repetitive behaviours, social interaction, social communication and emotional responses subscales compared with each of the other time points (T1-Baseline, T2-Control and T4-Lockdown). On the other hand, the cognitive style and maladaptive speech subscales did not show significant differences in any of the measurements. It should be noted that a low score indicates a decrease in the severity of the characteristics of children with ASD. Furthermore, a deterioration in the same subscales was observed following the 8-week COVID-19 lockdown period at T4-Lockdown compared to the other time points (T1-Baseline, T2-Control and T3-Judo). No significant differences in any of the subscales were observed during the control period between T1-Baseline and T2-Control.

## 4. Discussion

This study confirms the positive effects of 8-week adapted judo training in repetitive behaviours, social interaction, social communication and emotional responses subscales in children with ASD. Furthermore, our results provide evidence of the indirect, potentially harmful effects on children with ASD during the lockdown imposed by the health authorities to stop the spread of COVID-19. The interpretation and limitations of the present study must be considered with respect to the exceptionality of the situation caused by the COVID-19 pandemic which altered the original intervention. This situation has caused an interruption in their daily routines leading to the increased time spent engaging in sedentary activities such as watching television and using electronic devices [35]. Under these circumstances, children with ASD risk losing the benefits they may have previously attained from an active lifestyle, which is demonstrated by the GARS-3 subscale scores measured in excess of baseline and control values prior to the improvements conferred by the adapted judo programme.

The overall results show that the participants recorded significantly better scores (*p* < 0.05) on the subscales repetitive behaviours, social interaction, social communication and emotional responses during the adapted judo intervention period than they did during the control period or during the lockdown. The first of the subscales, repetitive behaviours, measures the restrictive/repetitive behaviours displayed by children with ASD. It corresponds to the subscale on stereotyped behaviours in the previous version of the testing instrument (GARS-2). The most common of these behaviours include a rocking motion of the hands, nodding, shaking arms, sudden running, rocking the body forward and backward, repeated manipulation of objects and finger movements [36]. Our results agree with those obtained by the Ferreira et al. [37] meta-analysis that reported children with ASD showed 1.1 fewer instances of stereotyped behaviours after an intervention with physical exercise. This clear indication of the effectiveness of physical activity for children with ASD has also been demonstrated following a variety of interventions, including an eight-week sports programme based on exercise with a ball [13], a ten-week horseback riding programme [38] and a programme linking physical exercise to video games [39].

The results of this study also, in essence, confirm prior research examining the use of combat sports or martial arts to improve certain executive and psychosocial behaviours that also influence the quality of life of children with ASD. For example, an intervention with adapted mixed martial arts [40] found improvements in executive functioning and reduced repetitive behaviours. These researchers attribute the improvements to the ability of these sporting activities to be adapted to the individual needs, preferences, and training status of children with ASD. Bahrami et al. [20] also achieved positive results in terms of reduced stereotypical behaviours following a karate kata intervention with highly structured activities calling for participants to follow a classmate or instructor’s movements and imagine scenarios with opponents. The success of our adapted judo programme could be partially due to its grounding in traditional martial arts practices, which, in addition to developing physical skills, aim to hone participants’ self-discipline and enhance their behavioural, emotional and cognitive control [21]. These psychosociological benefits align with the recommendations of the most successful martial arts interventions for children with ASD [20,40].

The social interaction, social communication and emotional responses subscales, which are closely connected to social ability, also displayed significant improvements following the judo intervention over the scores for the control period. These results support and confirm previous findings showing that participation in sport can improve the social abilities of children with ASD and help improve engagement in social interactions [15,16]. Sports programmes [13,14] are ideal for fostering positive social ability since the very act of participation and need for teamwork involved afford children innumerable opportunities to interact with one another. The current findings coincide with Movahedi et al. [19], which showed socio-emotional improvements after a karate intervention that requires participants to engage with their surroundings and with one another. In our research, touch or contact is our most social sense since it involves exploring the environment and engaging in successful interactions, forming interpersonal attachments. In that regard, judo practice involves physical contact situations during standing and groundwork, and contact situations have a potential role in developing bodily self-awareness defined as the ability to sense and recognise our body as our own [41]. The information that arises from inside the body gives information about its movement and location in the space (e.g., proprioceptive, vestibular and kinaesthetic input) and the perception of its physiological condition [41]. Thus, our adapted judo programme introduced a progressive increase in oppositional situations where physical contact is promoted between peers in simple to increasingly complex situations, thus eliciting progressive training of perception and decision-making skills. Contact experienced by the participants during the adapted judo programme may have stimulated positive adaptations in self-awareness in addition to the enhancement obtained in behavioural, social and emotional skills. Future works are warranted to investigate the effects of judo practice on self-awareness in children with ASD.

Cognitive style and maladaptive speech did not present differences during the lockdown period or the adapted judo programme. Nevertheless, some physical activity interventions have shown improvements in cognitive functioning [39,42], likely because the control of bodily movement developed in the context of sport involves decision making, anticipation and the measuring of speed and trajectories, all of which are associated with cognitive skills [43]. Nonetheless, cognitive improvements and decreases in inappropriate language are less positively affected by interventions with minimal physical activity compared to social abilities [44]. For example, Pan et al. [43] found significant improvements on only three of the six indices of the Wisconsin Card Sorting Test, while Anderson-Hanley et al. [39] found significant progress on the Digit Span Backward Task, but not the Color Trails or Stroop Tests. These findings suggest that the two subscales in question might not be sensitive to participation in an adapted judo programme while potentially being more connected to other symptoms within the broad spectrum of autism.

The results following the COVID-19 lockdown showed a significant, generalised decline in the same behavioural measures (repetitive behaviours, social interaction, social communication and emotional responses) that were improved with the adapted judo programme in children with ASD. The quarantine likely resulted in additional stress among parents and damage to the mental health of children with ASD, leading to a greater number of episodes of aggression and maladjusted behaviour [45]. According to Narzisi [46], during the COVID-19 lockdown period, children with ASD may have an increased tendency toward stereotyped behaviour because of the new stress caused by changes in the daily routine. Meanwhile, it is also possible that the deterioration in behaviours was due to the lockdown and the halt of the adapted judo programme, as in this study, it is impossible to separate these two factors. The fact that the baseline (T1) and control scores (T2) were lower than the results after eight weeks of the lockdown agrees with other recent studies of children with ASD [47] that reported worsened sleep problems and autism symptoms during the lockdown. These factors may be compounded by the preference of individuals with ASD for sedentary activities, especially those involving screens [2], which can increase their risk of obesity and cardiovascular disease [10].

In order to counteract these harmful outcomes, it would be beneficial for children with ASD to continue engaging in physical activity interventions at home during quarantine orders to maintain overall health and immune function while minimising sedentary screen time [45]. Narzisi [46] recommends that parents and children share activities and play semi-structured games together or do exercise routines using online videos; however, there is an inherent need for familial support for these efforts to be successful. Furthermore, there is a wealth of resources and articles that offer examples of physical activities to do with children with ASD and strategies to encourage them to be more active [35]. As part of our project, we have created online judo materials to offer the study participants the opportunity to receive judo instruction remotely.

The limitations of this study stem from the small sample size. However, it is worth emphasising the difficulties involved in taking longitudinal measurements of a group of children with ASD due to their high drop-out rate and their tendency to display a low degree of continuity in physical exercise. Because of the difficulties discussed above, we could not recruit a control group, representing another limitation of the study. However, we established a control period, during which the participants did not do any extracurricular sports that also encompassed their winter break from classes. Finally, the fact that the parents or guardians completed the questionnaires could have affected the results. While they were certainly in the best position to follow the participants’ progress, they may also have been influenced by their expectations of the adapted judo programme or the stress they experienced during the lockdown. Therefore, these expectations force us to consider these data as preliminary results that require a more extended period of intervention to allow for broader interpretation.

## 5. Conclusions

The eight-week adapted judo training intervention positively affected repetitive behaviours, social interaction, social communication, and emotional responses subscales in children with ASD. On the other hand, the cognitive style and maladaptive speech subscales did not show significant differences in any of the measurements. The COVID-19 lockdown period resulted in an apparent deterioration of repetitive behaviour, social interaction, social communication and emotional response returning to baseline values in children with ASD. The cognitive style and maladaptive speech subscales did not display any changes over any of the periods. Further studies could attempt to replicate or expand these findings either in-person or remotely and incorporate factors that influence cognition or language into the judo programme. The long-term effects of these interventions also need to be explored, while additional aspects of behaviour, including issues connected to motor skills, should be examined.

## Figures and Tables

**Figure 1 ijerph-18-08515-f001:**
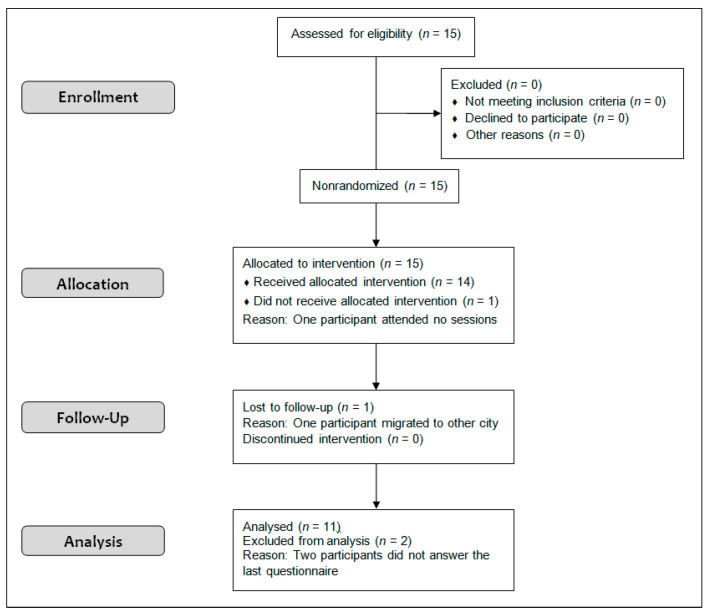
Flow chart of the Transparent Reporting of Evaluations with Non-randomized Designs (TREND) shows the number of participants through each stage of the study.

**Figure 2 ijerph-18-08515-f002:**
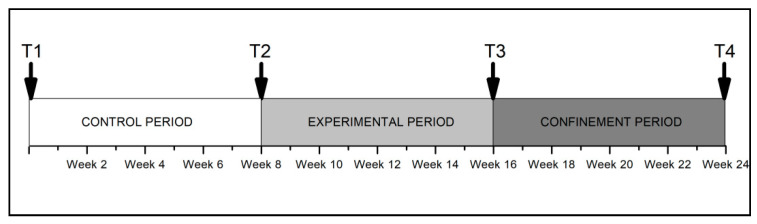
Study timeline. All participants were assessed four times: once as a baseline measurement upon entry to the programme (T1-Baseline), a second time after an eight-week control period (T2-Control), a third time after an eight-week adapted judo intervention (T3-Judo) and a fourth time after an eight-week lockdown period due to COVID-19 (T4-Lockdown).

**Figure 3 ijerph-18-08515-f003:**
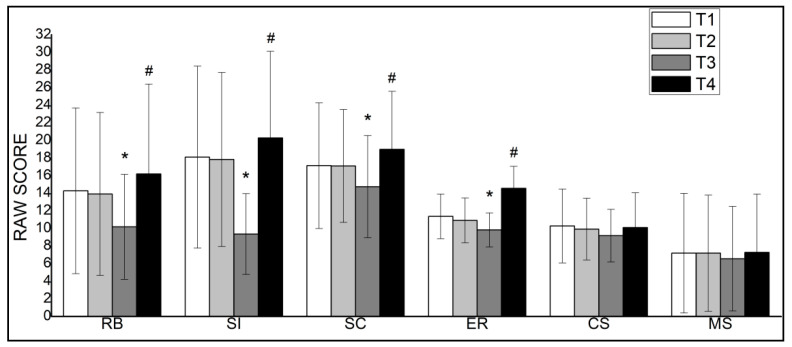
Gilliam Autism Rating Scale-Third Edition (GARS-3) subscales for repetitive behaviours (RB), social interaction (SI), social communication (SC), emotional responses (ER), cognitive style (CS), and maladaptive speech (MS) at baseline (T1-Baseline), after the 8-week control period (T2-Control), after the 8-week judo training intervention (T3-Judo), and after the 8-week COVID-19 lockdown period (T4-Lockdown). * significantly different (*p* < 0.05) from T1-Baseline, T2-Control, and T4-Lockdown. # significantly different (*p* < 0.05) from T1-Baseline, T2-Control, and T3-Judo.

## Data Availability

All data files are available from the FIGSHARE database: https://figshare.com/s/2943e0ef5369efcbf48f (accessed date on 30 June 2020).

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
