# Peer review of "Behavioural Improvements in Children with Autism Spectrum Disorder after Participation in an Adapted Judo Programme Followed by Deleterious Effects during the COVID-19 Lockdown"

_ijerph, 2021, doi:10.3390/ijerph18168515_

Round 1

Reviewer 1 Report

This manuscript presents some interesting findings, and it is unfortunate that the study was interrupted by the COVID-19 pandemic.  However, the authors have been resourceful in their approach and have ensured that the research remains useful.  There are a few main issues with the study: (1) the sample size is very small, (2) there is no control group, and (3) although the effects observed could be attributable to the intervention and to lockdown, there are other possibilities that cannot be ruled out.  Each of these issues have been recognised and addressed (albeit briefly) in the manuscript.  The authors do a very good job in not overstating the findings, though it might be worth making it clearer that, due to the above issues, this study should be treated very much as a pilot, with the preliminary findings presented here perhaps paving the way for more rigorous research on the topic.  The manuscript itself is very nicely written, although I did spot a few typos here and there.

Abstract

  • It would be useful to note what behavioural and psychosocial variables were being considered here.

Introduction

  • Could training in martial arts not also be helpful in building confidence and perhaps reducing stress and anxious/depressive symptoms in autistic children?

Methods

  • It is mentioned that the ERASMUS + Sport programme covered six countries: are similar data available from the other sites, and if so, could they be considered/presented here for higher powered analyses?
  • 150: “data was” should be “data were”.
  • It would be useful to report the dates at which each of the stages of the study took place (i.e., so that readers can refer to these in relation to where the world [and in particular, Spain] was at in terms of the pandemic, lockdowns etc.)
  • 205: Please remove the space before the comma preceding “emotional responses”.
  • As there was no control group, there may be expectation effects, and parents rather than the children completed the questionnaires, it is difficult to know whether the effects observed are attributable to the intervention. I note that these issues are mentioned in the Discussion, but only fairly briefly.

Results

  • It would be helpful to mention which subscales did not show statistically significant differences within the text rather than only in relation to Figure 3.
  • I’d advise removing the power calculation because post hoc power analyses like this will always show high statistical power if the observed effects are statistically significant (i.e., because the power is directly related to the p value) (please see the article by Hoenig & Heisey, 2001). A power calculation based on these data would therefore arguably be more useful for determining the minimum required sample size for a follow-up study.

Discussion

  • “our results provide evidence of the effects of the lockdown measures taken by the health authorities with the aim of avoiding the spread of COVID-19 on this population.” It is arguably difficult to attribute these observations to any specific cause.  As there have been so many things going on regarding COVID-19 (e.g., school closures, parental job insecurities, longer waiting lists for medical care, general uncertainty, heightened anxieties in others), it is arguably not justified to attribute this effect specifically to lockdown (though it may of course have played a significant role).
  • 292: Please remove unnecessary “the” before “these sporting activities”.
  • 305: Please remove “that” before “showing that”.
  • 315: Please remove “which” before “are related”.
  • Some consideration of how sensory hypersensitivities could be relevant in the context of contact sports such as judo may be useful here.
  • Some discussion of the possible risks to autistic children taking part in contact sports such as judo could be considered. For instance, this group may be a greater risk due to increased likelihood of co-occurring dyspraxia/motor coordination difficulties.  Also, although it may go beyond the remit of the current paper, I did wonder what the implications might be for autistic children who do not have access to such interventions but may benefit from taking part.  For instance, is there any advice to parents as to whether autistic children should take part in general population judo classes etc., as this might be considerably more challenging/risky etc.?
  • 338: Please remove “higher the” before “(T1)”.
  • 338-341: Please check direction of effect here, as higher scores appear to relate to poorer outcomes (also, please add “in” before “accordance”).
  • 351: Please add a full stop before “Furthermore”.

Acknowledgements

  • 400-401: “thanks” should be “thank”, and “his” should be “their”.
  • There is a bracket missing at the end of the last sentence of this section.

References

Hoenig, J. M., & Heisey, D. M. (2001). The abuse of power: The pervasive fallacy of power calculations for data analysis. The American Statistician, 55(1), 19–24. https://doi.org/10.1198/000313001300339897

Author Response

RESPONSE TO REVIEWERS

REVIEWER 1

AUTHORS RESPONSE: The authors would like to thank the reviewer for valuable comments that have contributed to the improvement of the manuscript.

Reviewer #1: This manuscript presents some interesting findings, and it is unfortunate that the study was interrupted by the COVID-19 pandemic.  However, the authors have been resourceful in their approach and have ensured that the research remains useful.  There are a few main issues with the study: (1) the sample size is very small, (2) there is no control group, and (3) although the effects observed could be attributable to the intervention and to lockdown, there are other possibilities that cannot be ruled out.  Each of these issues have been recognised and addressed (albeit briefly) in the manuscript.  The authors do a very good job in not overstating the findings, though it might be worth making it clearer that, due to the above issues, this study should be treated very much as a pilot, with the preliminary findings presented here perhaps paving the way for more rigorous research on the topic.  The manuscript itself is very nicely written, although I did spot a few typos here and there.

AUTHORS RESPONSE: The reviewer is correct. The COVID lockdown happened suddenly and interrupted a project in which we had invested a lot of time and effort. Fortunately, we were able to use some of the data and redirect the research to observe the effects of the lockdown. The circumstances allowed us to include 8-week periods of control, intervention and lockdown with the same group of participants. We are going to include some paragraphs at the beginning of the discussion reinforcing the idea that the results are preliminary and reflect an exceptional situation and should be interpreted as a pilot study of subsequent applications. We are clear about the limitations of the manuscript, such as the number of participants and the presence of a control group, it is really difficult to meet these conditions with this type of population. Due to the exceptional situation experienced due to the covid-19 pandemic, we believe it is interesting to share the subsequent results caused by the lockdown that can serve as a sample of the impact it has on children with ASD.

We corrected typos and restructured the abstract

Abstract

Reviewer #1: It would be useful to note what behavioural and psychosocial variables were being considered here.

AUTHORS RESPONSE: Our intention was to explain in the abstract what types of behaviors had been analyzed: repetitive behaviors, social interaction, social communication, emotional responses, cognitive style and maladaptive speech. Due to the word limit they were not entered. We have currently adapted the abstract.

Introduction

Reviewer #1: Could training in martial arts not also be helpful in building confidence and perhaps reducing stress and anxious/depressive symptoms in autistic children?

AUTHORS RESPONSE: We have incorporated a text and some references in the introduction reinforcing the reviewer's suggestion

Methods

Reviewer #1: It is mentioned that the ERASMUS + Sport programme covered six countries: are similar data available from the other sites, and if so, could they be considered/presented here for higher powered analyses?

AUTHORS RESPONSE: We have added this text….”This same program is carried out in six countries of the European Union, due to the fact that the data are not homogeneous in all the countries due to different dates in the lockdown, the lack of control period and different intervention times. It is appropriate to include only the data from one of the countries, in this sense we have put rigor and control in the data collection before a larger sample”.

Reviewer #1: 150: “data was” should be “data were”.

AUTHORS RESPONSE: This has been corrected.

Reviewer #1: It would be useful to report the dates at which each of the stages of the study took place (i.e., so that readers can refer to these in relation to where the world [and in particular, Spain] was at in terms of the pandemic, lockdowns etc.)

AUTHORS RESPONSE: We have added the exact dates of the entire process

Reviewer #1: 205: Please remove the space before the comma preceding “emotional responses”.

AUTHORS RESPONSE: This has been corrected.

Reviewer #1: As there was no control group, there may be expectation effects, and parents rather than the children completed the questionnaires, it is difficult to know whether the effects observed are attributable to the intervention. I note that these issues are mentioned in the Discussion, but only fairly briefly.

AUTHORS RESPONSE: We have taken this suggestion into account in the discussion

Results

Reviewer #1: It would be helpful to mention which subscales did not show statistically significant differences within the text rather than only in relation to Figure 3.

AUTHORS RESPONSE: It has been reported in the text of the subscales that they did not have significant differences

Reviewer #1: I’d advise removing the power calculation because post hoc power analyses like this will always show high statistical power if the observed effects are statistically significant (i.e., because the power is directly related to the p value) (please see the article by Hoenig & Heisey, 2001). A power calculation based on these data would therefore arguably be more useful for determining the minimum required sample size for a follow-up study.

AUTHORS RESPONSE: We have removed this information from the text. We did not know this paper and it is very interesting, we greatly appreciate this suggestion.

Discussion

Reviewer #1: “our results provide evidence of the effects of the lockdown measures taken by the health authorities with the aim of avoiding the spread of COVID-19 on this population.” It is arguably difficult to attribute these observations to any specific cause.  As there have been so many things going on regarding COVID-19 (e.g., school closures, parental job insecurities, longer waiting lists for medical care, general uncertainty, heightened anxieties in others), it is arguably not justified to attribute this effect specifically to lockdown (though it may of course have played a significant role).

AUTHORS RESPONSE: It is evident that a specific cause cannot be attributed to the deterioration in the behaviors of children with ASD, for this reason we have softened the forcefulness of this statement by explaining the characteristics of lockdown:….”our results provide evidence of the indirect effects related to the effects of the lockdown measures taken by the health authorities with the aim of avoiding the spread of COVID-19 on this population. This situation has caused an interruption in their daily routines leading to an increased amount of time spent engaging in sedentary activities such as watching television and using electronic devices….”

Reviewer #1: 292: Please remove unnecessary “the” before “these sporting activities”.

AUTHORS RESPONSE: This has been corrected.

Reviewer #1: 305: Please remove “that” before “showing that”.

AUTHORS RESPONSE: This has been corrected.

Reviewer #1: 315: Please remove “which” before “are related”.

AUTHORS RESPONSE: This has been corrected.

Reviewer #1: Some consideration of how sensory hypersensitivities could be relevant in the context of contact sports such as judo may be useful here.

AUTHORS RESPONSE: As described in the intervention section, a specially designed adaptation of judo teaching for children with ASD is applied. It begins with simple elements of repetition and imitation and continues with the collaboration of a partner. The fact of practicing with the judogi (traditional judo uniform) facilitates the work in pairs, since the actions are not carried out directly on the body of the partner, but through the judogi, which serves as a transmission instrument to apply all the actions. This particularity of judo can help children with hypersensitivity and rejection of contact.

Reviewer #1: Some discussion of the possible risks to autistic children taking part in contact sports such as judo could be considered. For instance, this group may be a greater risk due to increased likelihood of co-occurring dyspraxia/motor coordination difficulties.  Also, although it may go beyond the remit of the current paper, I did wonder what the implications might be for autistic children who do not have access to such interventions but may benefit from taking part.  For instance, is there any advice to parents as to whether autistic children should take part in general population judo classes etc., as this might be considerably more challenging/risky etc.?

AUTHORS RESPONSE: According to the experience we have so far, the risk of practicing adapted judo is practically nil, since none of the centers where the program is carried out has any injury been reported. In case of dyspraxia or motor coordination difficulties, they are considered in the learning rhythm of each individual, therefore each child advances according to their motor possibilities. The beginning of the program is based on actions of imitation and repetition, with a progressive increase in oppositional situations where perception and decision-making is stimulated in simple to increasingly complex situations. The fact that judo is a grip-based sport facilitates personal interaction, in addition the fact that grip occurs through the judo uniform facilitates the control of the partner and the application of forces, since they are not performed directly on the partner's body. As you advance in the technical mastery, more hand-to-hand interactions occur. The following paragraph has been inserted within the manuscript to explain the likely contact effects:

  • Page 8, lines 849-865: “Paradoxically, touch or contact is our most social sense since it involves exploring the environment and engaging in successful interactions, forming interpersonal attachments. In that regard, Judo practice involves contact situations during standing and groundwork, and physical contact situations have a potential role in developing bodily self-awareness defined as the ability to sense and recognise our body as our own [NUMBER]. The information that arises from inside the body gives in-formation about its movement and location in the space (e.g., proprioceptive, vestibular and kinaesthetic input) and the perception of its physiological condition [NUMBER]. Thus, our adapted judo program introduced a progressive increase in oppositional situations where physical contact is promoted between peers in simple to increasingly complex situations, thus eliciting progressive training of perception and decision-making skills. Contact experienced by the participants during the adapted judo programme may have stimulated positive adaptations in self-awareness in addition to the enhancement obtained in behaviour, social and emotional skills. Future works are warranted to investigate the effects of judo practice in self-awareness in children with ASD.

Reviewer #1: 338: Please remove “higher the” before “(T1)”.

AUTHORS RESPONSE: This has been corrected.

Reviewer #1: 338-341: Please check direction of effect here, as higher scores appear to relate to poorer outcomes (also, please add “in” before “accordance”).

AUTHORS RESPONSE: This has been corrected. It was a mistake, because it should be noted that a low score indicates a decrease in the severity of the characteristics of children with ASD.

Reviewer #1: 351: Please add a full stop before “Furthermore”.

AUTHORS RESPONSE: This has been added.

Acknowledgements

Reviewer #1: 400-401: “thanks” should be “thank”, and “his” should be “their”.

AUTHORS RESPONSE: This has been corrected.

Reviewer #1: There is a bracket missing at the end of the last sentence of this section.

AUTHORS RESPONSE: This has been added.

References

Reviewer #1: Hoenig, J. M., & Heisey, D. M. (2001). The abuse of power: The pervasive fallacy of power calculations for data analysis. The American Statistician55(1), 19–24. https://doi.org/10.1198/000313001300339897

Reviewer 2 Report

The main purpose of the study was to compare the psychosocial and behavioural scores of children with Autism Spectrum Disorder before their participation in an adapted judo programme.

It is a study focused on a very special population, well conducted and with a correct design. Some considerations will be made for a better understanding and reading of this work.

1. In general, a revision of the formal aspects of the text is recommended to improve some typographical details (for example, the end of lines 52, 163, 274, etc.).

2. It is recommended that the title include the term Autism Spectrum Disorder, for a better clarification of the content.

3. It is also recommended that the term Autism Spectrum Disorder prior to ASD be included in the abstract (although the acronyms are widely recognized in the medical and scientific literature).

4. It is recommended that in the keywords, at least, the word Autism is included, for a better search in the databases.

5. The study actually involves a small sample (11 children). We consider that it is not a limitation taking into account the target population to which it is directed, and the design considerations, so, in this case, it does not mean undervaluing the results.

6. Statistical analysis and selected tests make it possible to guarantee the value of the results.

8. Authors are recommended to indicate whether they performed "free practice", that is/or, tasks with decisional content, in the intervention phase. If not, explain why. Interaction through tasks with decisional content (and not exclusively psychomotor) could even improve the results in this specific population.

7. In the Discussion section, the authors clarify the value of sport, and especially the value of martial arts (for example, using kata in karate). In this section, the authors are recommended to state, in their opinion, the reasons why a sport with direct interaction, and mediated through a grip, (in addition to groundwork) could have an even greater impact on the improvements of the "social relations" of this population; that is to say, justify the additional benefits that the use of motor skills and the specific coordination demands of judo, as a sport, can bring.

8. Finally, in the Conclusions section, the authors declare the impact of the COVID-19 lockdown on the deterioration of certain values ​​in children with ASD. It is recommended that the authors also present in this section the benefits found and associated (on those same values) to an intervention based on the sport of Judo.

Author Response

REVIEWER 2

Reviewer #2: The main purpose of the study was to compare the psychosocial and behavioural scores of children with Autism Spectrum Disorder before their participation in an adapted judo programme.

It is a study focused on a very special population, well conducted and with a correct design. Some considerations will be made for a better understanding and reading of this work.

AUTHORS RESPONSE: The authors would like to thank the reviewer for valuable comments that surely have contributed to the improvement of the manuscript.

Reviewer #2: 1. In general, a revision of the formal aspects of the text is recommended to improve some typographical details (for example, the end of lines 52, 163, 274, etc.).

AUTHORS RESPONSE: The authors appreciate that the reviewer has identified and reported typographical errors and apologize for this oversight. In the process of reviewing typographical errors, we have corrected the errors and indicate them in the text turning on the tracked changes tool. Furthermore, we have made several changes to improve the clarity and readability of the document.

Reviewer #2: 2. It is recommended that the title include the term Autism Spectrum Disorder, for a better clarification of the content.

AUTHORS RESPONSE: The term Autism Spectrum Disorder was included in the title.

Reviewer #2: 3. It is also recommended that the term Autism Spectrum Disorder prior to ASD be included in the abstract (although the acronyms are widely recognized in the medical and scientific literature).

AUTHORS RESPONSE: The term Autism Spectrum Disorder was included in the abstract.

Reviewer #2: 4. It is recommended that in the keywords, at least, the word Autism is included, for a better search in the databases.

AUTHORS RESPONSE: Autism was added to keywords.

Reviewer #2: 5. The study actually involves a small sample (11 children). We consider that it is not a limitation taking into account the target population to which it is directed, and the design considerations, so, in this case, it does not mean undervaluing the results.

AUTHORS RESPONSE: Thank you for your consideration.

Reviewer #2: 6. Statistical analysis and selected tests make it possible to guarantee the value of the results.

AUTHORS RESPONSE: Thank you for your comment.

Reviewer #2: 8. Authors are recommended to indicate whether they performed "free practice", that is/or, tasks with decisional content, in the intervention phase. If not, explain why. Interaction through tasks with decisional content (and not exclusively psychomotor) could even improve the results in this specific population.

AUTHORS RESPONSE: Teachers introduced opposition games progressively, but it was always a practice conditioned to a simple objective (to get out of space, touch a part of the opponent's body ...). Children with ASD have dyspraxia and motor coordination difficulties, and they struggle to understand very complex tasks or with little defined objectives. No traditional  "free practice" was utilized, although opposition situations with training partners of similar physical and cognitive levels were introduced. In these situations, contact was gradually introduced, and the complexity of the stimuli they had to attend to, and the decisions they had to make, increased very gradually. We have added a paragraph in the manuscript to refer to this: 

  • Page 8, lines 858-865: “Thus, our adapted judo program introduced a progressive increase in oppositional situations where contact is promoted between peers in simple to increasingly complex situations, thus ensuing perception and decision-making skills progressive training. Contact experienced by the participants during the adapted judo programme may have stimulated positive adaptations in self-awareness in addition to the enhancement obtained in behaviour, social and emotional skills. Future works are warranted to investigate the effects of judo practice in self-awareness in children with ASD.”

Reviewer #2: 7. In the Discussion section, the authors clarify the value of sport, and especially the value of martial arts (for example, using kata in karate). In this section, the authors are recommended to state, in their opinion, the reasons why a sport with direct interaction, and mediated through a grip, (in addition to groundwork) could have an even greater impact on the improvements of the "social relations" of this population; that is to say, justify the additional benefits that the use of motor skills and the specific coordination demands of judo, as a sport, can bring.

AUTHORS RESPONSE: We have inserted in the discussion the following paragraph to attend to the recommendation of the reviewer:

  • Page 8, Lines 849-858. “Paradoxically, touch or contact is our most social sense since it involves exploring the environment and engaging in successful interactions, forming interpersonal attachments. In that regard, Judo practice involves physical contact situations during standing and groundwork, and contact situations have a potential role in developing bodily self-awareness defined as the ability to sense and recognise our body as our own [NUMBER]. The information that arises from inside the body gives information about its movement and location in the space (e.g., proprioceptive, vestibular and kinaesthetic input) and the perception of its physiological condition [NUMBER].”

Reviewer #2: 8. Finally, in the Conclusions section, the authors declare the impact of the COVID-19 lockdown on the deterioration of certain values ​​in children with ASD. It is recommended that the authors also present in this section the benefits found and associated (on those same values) to an intervention based on the sport of Judo.

AUTHORS RESPONSE: We have added the text below these lines to the conclusion section. Thanks to the reviewer for their comment as we really believe that they give a higher value to the article.

Page 9, lines 980-983. “The eight-week adapted judo training intervention positively affected repetitive behaviours, social interaction, social communication, and emotional responses subscales in children with ASD. On the other hand, the cognitive style and maladaptive speech subscales did not show significant differences in any of the measurements.”

Round 2

Reviewer 1 Report

The authors have clearly considered my suggestions and amended their manuscript accordingly.  I think it is now a stronger paper and recommend that it be published.  I have noted below a few very minor suggestions and pointed out a few places where I noticed small typos/grammatical mistakes.

  • Perhaps reorder the first paragraph of the Discussion so that it presents the main findings first and then explains how these should be considered in relation to lockdown etc.?
  • Please check spelling used for programme/program is consistent throughout.
  • 24: “across four time-points” rather than “between four times”?
  • 28: If the word limit allows for it, please specify which variables showed significant improvements.
  • 56: “should be “Participating in sports is” (rather than “are”).
  • 111: Perhaps change “slated to run” to “scheduled to run”?
  • 140: Please include mean and SD here if available.
  • 150/Figure 1: Looks like there’s an unnecessary space after “Excluded”. Also, in “Analysis” section, please change “didn’t” to “did not”.
  • 157: “23th” should be “23rd”.
  • 230-231: This should be “Shapiro-Wilk test”.
  • 250: Perhaps change to “each of the other time points”?
  • 288-290: The text does not need to be bold here.
  • 361: No need to start a new paragraph here (also generally better to avoid single sentence paragraphs anyway).
  • 377-378: Please check the grammar here.
  • 422: No need for brackets around “Club Judo Louis”.
  • 423: As it appears that the authors are referring to an organisation rather than an individual here, there are two instances of “his” that should be amended to “their”.

Author Response

RESPONSE TO REVIEWERS

REVIEWER 1

Reviewer #1: The authors have clearly considered my suggestions and amended their manuscript accordingly.  I think it is now a stronger paper and recommend that it be published.  I have noted below a few very minor suggestions and pointed out a few places where I noticed small typos/grammatical mistakes.

AUTHORS’ RESPONSE: The authors wish to thank the reviewer for the second round of review that has contributed to the improvement of the manuscript.

Reviewer #1: Perhaps reorder the first paragraph of the Discussion so that it presents the main findings first and then explains how these should be considered in relation to lockdown etc.?

AUTHORS’ RESPONSE: We have reordered the first paragraph of the discussion with the reviewer's suggestions.

Reviewer #1: Please check spelling used for programme/program is consistent throughout.

AUTHORS’ RESPONSE: We have searched for all the words written as "program" and have replaced them with "program". We would like to apologize for this lack of rigour, often caused by the submission deadline.

Reviewer #1: 24: “across four time-points” rather than “between four times”?

AUTHORS’ RESPONSE: This has been changed.

Reviewer #1: 28: If the word limit allows for it, please specify which variables showed significant improvements.

AUTHORS’ RESPONSE: The number of words fits.

Reviewer #1: 56: “should be “Participating in sports is” (rather than “are”).

AUTHORS’ RESPONSE: This has been corrected.

Reviewer #1: 111: Perhaps change “slated to run” to “scheduled to run”?

AUTHORS’ RESPONSE: This has been changed.

Reviewer #1: 140: Please include mean and SD here if available.

AUTHORS’ RESPONSE: We have added the mean and standard deviation.

Reviewer #1: 150/Figure 1: Looks like there’s an unnecessary space after “Excluded”. Also, in “Analysis” section, please change “didn’t” to “did not”.

AUTHORS’ RESPONSE: This has been corrected.

Reviewer #1: 157: “23th” should be “23rd”.

AUTHORS’ RESPONSE: This has been corrected.

Reviewer #1: 230-231: This should be “Shapiro-Wilk test”.

AUTHORS’ RESPONSE: This has been corrected.

Reviewer #1: 250: Perhaps change to “each of the other time points”?

AUTHORS’ RESPONSE: This has been changed.

Reviewer #1: 288-290: The text does not need to be bold here.

AUTHORS’ RESPONSE: This has been changed.

Reviewer #1: 361: No need to start a new paragraph here (also generally better to avoid single sentence paragraphs anyway).

AUTHORS’ RESPONSE: We have attached it to the previous paragraph

Reviewer #1: 377-378: Please check the grammar here.

AUTHORS’ RESPONSE: We have deleted a sentence as it was not necessary and did not make sense.

Reviewer #1: 422: No need for brackets around “Club Judo Louis”.

AUTHORS’ RESPONSE: This has been changed.

Reviewer #1: 423: As it appears that the authors are referring to an organisation rather than an individual here, there are two instances of “his” that should be amended to “their”.

AUTHORS’ RESPONSE: This has been corrected.